# Development of a Rapid Diagnostic Kit for Congestive Heart Failure Using Recombinant NT-proBNP Antigen

**DOI:** 10.3390/medicina57080751

**Published:** 2021-07-25

**Authors:** Young-Ki Lee, Dong-Ok Choi, Ga-Yeon Kim

**Affiliations:** 1Department of Biomedical Laboratory Science, College of Health Sciences, Dankook University, 119 Dandae-ro, Dongnan-gu, Cheonan-si 31116, Chungnam, Korea; pp99pp@dankook.ac.kr; 2Bore Da BioTECH Co., LTD., 14, B-505, Sagimakgol-ro 45 beon-gil, Jungwon-gu, Seongnam-si 13209, Gyeonggi-do, Korea; realchoi@boreda.com; 3Department of Public Health, Graduate School, Dankook University, 119 Dandae-ro, Dongnan-gu, Cheonan-si 31116, Chungnam, Korea

**Keywords:** BNP, congestive heart failure, NT-proBNP, rapid diagnostic kit, biomarker

## Abstract

*Background and Objectives:* In patients with congestive heart failure, brain natriuretic peptide (BNP) and N-terminal prohormone of brain natriuretic peptide (NT-proBNP) are released due to excessive heart muscle expansion; they can be used for the early detection, progress monitoring, and treatment of congestive heart failure. Recently, considerable efforts have been made to develop an NT-proBNP-based biomarker for detecting heart failure. This study attempts to develop a rapid and accurate congestive heart failure diagnostic kit using NT-proBNP. *Materials and Methods:* A new gene based on NT-proBNP was selected, recombined, and expressed in *Escherichia coli* strains, and then monoclonal antibodies were produced using the hybridoma technique. Additionally, antigen-antibody reactivity was confirmed using indirect enzyme-linked immunosorbent assay (ELISA). Furthermore, the first pair and full-strip pair tests were conducted to select candidate clones; these were applied to a rapid diagnosis kit based on gold conjugates and compared with other currently available antigens. *Results:* NT-proBNP-based antigens with high specificity and monoclonal antibodies were produced, and the optimal antigen-antibody reactivity was confirmed using indirect ELISA. The first pair and full-strip pair tests were performed to select the optimal candidate clones, and a rapid diagnosis kit with excellent reactivity was developed by applying these to a rapid diagnosis kit based on gold conjugates. *Conclusions:* The development of this rapid diagnosis kit with excellent performance in congestive heart failure is expected to improve disease management by providing an early assessment of the risk of heart failure.

## 1. Introduction

Congestive heart failure (CHF) is associated with high morbidity and mortality worldwide. Hence, early detection and treatment of this disease are of importance. Recently, considerable efforts, including clinical level studies, have been made to develop biomarkers for heart failure (HF) [1,2]. Types of biomarkers for measuring heart failure disease using peripheral blood include HF inflammation biomarkers (C reactive protein (CRP) and interleukin 6 (IL-6), HF myocyte injury biomarkers (cardiac-specific troponin I (CTnI), and creatine kinase-MB (CK-MB), and HF myocyte stress biomarkers (B-type natriuretic peptide (BNP) and ST2 proteins). Among them, BNP and ST2 proteins, which are myocardial stress peptide biomarkers, accurately reflect the worsening prognosis of HF patients. In particular, BNP is a strong prognostic factor for all causes of death in asymptomatic patients and heart failure patients at all disease stages [3]. The mechanism underlying the association between BNP and HF involves the downregulation of the renin-angiotensin-aldosterone system by BNP, which reduces the sympathetic nerve activity of the heart and kidneys. It increases blood flow to the kidneys as well as sodium secretion by directly acting on renal collecting ducts [4]. Similar to BNP, N-terminal prohormone of brain natriuretic peptide (NT-proBNP), a proBNP derivative, is a new and important biomarker determining the severity of a heart attack [5,6]. This biomarker comprises 108 amino acids and is produced in ventricular muscle cells (Figure 1). It is split by endoprotease furin into the active C-terminal polypeptide BNP, consisting of 32 amino acids, and inactive NT-proBNP, comprising 76 amino acids [7]. The secretion of these two peptides is increased by ventricular wall expansion and volume overload, resulting in the release of both at the same rate. Also, due to its longer half-life, NT-proBNP shows higher sensitivity to early stages of left ventricular disorders. Thus, the detection of NT-proBNP may improve HF prognosis and facilitate the effective management of disease by providing an accurate risk assessment of HF [8,9].

Although various laboratory tests enabling early detection of HF are available, these traditional diagnostic methods do not ensure the efficacy, accuracy, timeliness, or cost-effectiveness of clinical decisions, prognoses, and treatments. Thus, a new NT-proBNP biomarker-based test showing relatively high diagnostic and prognostic capabilities and clinical usefulness may fulfill the need for a rapid diagnostic method and point-of-care (POC) [10].

Cardiac resynchronization therapy (CRT) is highly effective for heart failure patients with type 2 diabetes mellitus, significantly improving clinical symptoms and reducing mortality and long-term morbidity [11,12]. A wide range of biomarkers can be used as predictors of the CRT response in heart failure patients receiving such treatments. BNT-proBNP, in particular, has high sensitivity and specificity, and there is a significant correlation between BNP levels and CRT results [13,14].

The objective of this study was to develop a rapid and accurate diagnostic kit for CHF, which can be applied in clinical practice by selecting and recombining a new gene based on NT-proBNP and expressing it in *Escherichia*
*coli.* This process develops an antigen with high specificity that produces hybridomas and monoclonal antibodies while identifying optimal antigen-antibody reactivity via indirect enzyme-linked immunosorbent assay (ELISA). A first pair test was conducted to select candidate clones, followed by selecting an optimal pair via a full-strip pair test. This pair was applied to a diagnosis kit based on gold conjugates to develop a rapid diagnosis kit based on NT-proBNP with excellent reactivity compared with other commercially available antigens. The results of this study are expected to significantly contribute to the early diagnosis and treatment of related diseases by reducing test costs and improving conventional methods that require expensive equipment and specialists.

## 2. Materials and Methods

### 2.1. NT-proBNP Gene Synthesis and Cloning

This study began in February 2018 and ended in October 2020 after receiving accreditation for animal testing by ethics review. The human proBNP sequence, registered in the National Center for Biotechnology Information (NCBI), was used as a reference to select a single segment of the amino acid sequence predicted to target and identify the NT-proBNP region (Table 1). Primer and gene synthesis, which incorporate sites for the HindIII and XhoI restriction enzymes in the selected amino acid sequence, were performed by Bioneer (Bioneer, Daejeon, Korea). The newly synthesized NT-proBNP was transformed into *E. coli* DH5α and incubated in LB broth (Sigma-Aldrich, Darmstadt, Germany) at 37 °C for 15 h. Plasmid DNA was extracted using a DNA extraction kit (Nucleogen, Seoul, Korea). Polymerase chain reaction (PCR) was performed using the extracted plasmid as a template. Primers containing a HindIII site in the forward region and an XhoI cleavage site in the reverse region (Table 2) were used to perform amplification under the following conditions: 94 °C for 30 s (denaturation), 53 °C for 30 s (annealing), and 72 °C for 30 s (extension). The entire process was repeated 37 times and allowed to react at 72 °C for 5 min in the post-extension period prior to completion. The PCR-amplified product was subjected to electrophoresis at 140 V on a 1.5% agarose gel for 20 min and stained with 0.5 mg/L of ethidium bromide (EtBr; 2%) for 30 min. The PCR products were observed using a Molecular Imager Gel DOC TM XR+ System (Bio-Rad, Irvine, CA, USA). Subsequently, the PCR-amplified product was eluted from agarose gel using a gel extraction kit (Nucleogen, Daejeon, Korea) for recombination. The recombinant mixture was prepared by adding 1 µL of the purified PCR product, 4 µL of distilled water, 1 µL of buffer A, 1 µL of buffer B, 2 µL of TA vector, and 1 µL of TA ligase to produce a final volume of 10 µL, and reacted at 25 °C for 2 h. The PCR product was recombined using a TA-cloning kit (Intron, USA). DNA transformation was achieved by combining the recombinant mixture with *E. coli* DH5α (stored at −80 °C) and stabilizing on ice for over 30 min. Subsequently, a heat block was prepared at 42 °C, and the preparation was subjected to heat shock for 1 min and 30 s. The transformed cell solution was plated on LB solid medium and cultured at 37 °C for 18 h. A single colony was randomly selected, inoculated again in 6 mL of LB medium, and cultured at 37 °C for 15 h. DNA was extracted using a DNA extraction kit (Bioneer, Daejeon, Korea).

### 2.2. Expression Vector Subcloning

In order to express the recombinant protein, subcloning was performed using the pET-21b vector (Novagen, Sigma-Aldrich, Darmstadt, Germany). After combining the expression vector and purified target inserts at a molar ratio of 1:3, 0.5 µL of T4 ligase and 1 µL of buffer were added to adjust the final volume to 10 µL and reacted at 37 °C for 1 h. In order to identify colonies containing inserts and the TA vector, colonies were treated with HindIII and XhoI. Subsequently, a single colony was selected and cultured on LB solid medium at 37 °C for 18 h. It was then reinoculated in 6 mL of LB medium and cultured at 37 °C for 15 h. DNA was extracted using a DNA extraction kit. Extracted DNA was added to 1 µL of enzyme buffer solution and 0.25 µL in each of the restriction enzymes, HindIII and XhoI, to adjust the final volume to 10 µL, and reacted at 37 °C for 1 h. Subsequently, electrophoresis in a 1.5% agarose gel was performed at 140 V for 20 min, followed by staining with EtBr (0.5 mg/L) for 30 min to identify the presence of the target fragment. Next, the confirmed DNA was transformed into *E. coli* BL21 (DE3) in the same manner as described above.

### 2.3. Recombinant Protein Expression and Purification

For protein expression, the transformed *E. coli* BL21 (DE3) strain was inoculated into 10 mL of LB medium, followed by seed culture at 37 °C for 15 h. Subsequently, 5 mL of seed culture solution, which was 1/100 of the final volume, was inoculated into 500 mL of LB medium, followed by incubation at 37 °C for 2 h. Following incubation, 0.25 mM isopropyl β-d-1-thiogalactopyrano-side (Goldbio, St. Louis, MO, USA) was added until the absorbance at 600 nm reached 0.6–0.8; afterward, protein expression was induced by shaking the culture at 180 rpm for 15 h at 18 °C [15]. Protein-expressing cells were then centrifuged in a 250-mL tube at 11,100× *g*, 4 °C for 10 min to precipitate cells. Phosphate-buffered saline (PBS, pH 7.4) was mixed with 1/10 of the culture volume solution and subjected to ultrasonic disintegration for 2 min. The supernatant was then recovered via centrifugation under conditions similar to those described above.

Ni-NTA affinity chromatography was used to purify the expressed recombinant proteins [16]. PBS (pH 7.4) was used as the binding buffer, whereas imidazole solution (adjusted to a concentration of 30 mM in PBS (pH 7.4) was used as the washing buffer; 250 mM imidazole buffer was used for elution. In order to remove the high concentration of imidazole in the elution, 5 L of PBS (pH 7.4) was prepared and maintained at 4 °C for 20 h for dialysis. Sodium dodecyl sulfate–polyacrylamide gel electrophoresis (SDS-PAGE) was performed using a vertical Mini-PROTEAN Tetra cell device (Bio-Rad, Irvine, CA, USA) to identify the protein. With the expected protein size considered, the concentration of acrylamide in the running gel and stacking gel was set to 14% and 4%, respectively. The size of the target protein was identified by Coomassie Blue staining (MilliporeSigma, Darmstadt, Germany) [17].

### 2.4. Mouse Immunization and Cell Fusion

Two 5-week-old (Balb/C) mice were immunized using the recombinant antigen. The experimental protocol was approved by the Institutional Animal Care and Use Review Committee of Bore Da BioTECH (BDIACUC20180221). After preparing the antigen at 1 mg/mL, an emulsion was prepared by mixing with complete Freund’s adjuvant (CFA; Sigma-Aldrich, Darmstadt, Germany) at a ratio of 1:1. The first immunization was performed on the sole of each mouse (100 µL). The second immunization was performed two weeks after the first immunization and then weekly, starting with the third immunization. From the second immunization to the fourth and final immunization, 75 µL was used. One week later, the mice were sacrificed by cervical dislocation, and the lymph nodes were removed. The extracted lymph nodes were fused with Sp2/O, a mouse myeloma cell line, using polyethylene glycol 1500 (Roche, Switzerland). To selectively culture the fused cells, hypoxanthine aminopterin thymidine (HAT, Sigma-Aldrich) medium was used. After preparing a 96-well plate and dispensing 200 µL/well, each plate was incubated at 37 °C in a CO_2_ incubator.

### 2.5. Hybridoma Screening

The cultured hybridomas were screened using an indirect ELISA. First, the recombinant antigen was dispensed into a 96-well plate at 50 µL/well and reacted at 4 °C for 16 h or longer. After washing with phosphate-buffered saline-Tween20 (PBS-Tween20, Merck), 0.25% casein (Sigma-Aldrich) was dispensed at 100 µL/well and reacted at 37 °C for 1 h or longer to block sites where the antigen was not attached. After washing with PBS-Tween20 (Merck), hybridoma culture medium was dispensed at 100 µL/well and incubated at 37 °C for 1 h. Samples were washed with PBS-Tween20 before each reaction. Horseradish peroxidase-conjugated goat anti-mouse IgG (Invitrogen, Irvine, CA, USA), diluted 1:10,000, was dispensed at 100 µL/well and incubated at 37 °C for 1 h. To detect the reaction, the color-developing substrate solution, 3,3′,5,5′-tetramethylbenzidine (Sigma-Aldrich), was dispensed at 50 µL/well to induce an enzymatic reaction. After reacting for 5 min, 1N H_2_SO_4_ was dispensed at 50 µL/well to terminate the enzymatic reaction. In order to determine the degree of color development, absorbance was measured at 450 nm using a microplate reader (Biotek, Winnooski, VT, USA), and 11 fused cells showing high absorbance were selected (Table 3).

### 2.6. Monoclonal Cell Selection and Culture

After dispensing the 11 selected cells into a 96-well plate, serial dilutions were performed. For selective culture, HAT medium was dispensed at 200 µL/well and cultured at 37 °C in a CO_2_ incubator. New HAT medium (100 µL/well) was added once every 2 days. Finally, seven candidate clones were selected by indirect ELISA (Table 4).

### 2.7. Ascites Production and Mass Production of Monoclonal Antibodies

The hybridomas selected for mass production of monoclonal antibodies were injected into the abdominal cavity of mice. When the mouse abdominal cavity swelled after 7 to 10 d, ascites were collected and centrifuged at 3243× *g*, 25 °C for 5 min. The supernatant was then collected and stored at −40 °C.

### 2.8. Purification of Monoclonal Antibodies

In order to purify monoclonal antibodies, ascites stored at −40 °C were dissolved at room temperature, diluted with PBS (pH 7.4) at a ratio of 1:3, and filtered through a 0.22-µm membrane filter. The proteins in this dialysate were subjected to A+G affinity chromatography and sufficiently washed with PBS (pH 7.4) for at least 30 min to remove nonspecific substances. The antibodies bound to the column were eluted with PBS (pH 3.2). At this time, 1 M neutralization buffer solution consisting of 1 M Tris-base (pH 9.0), which was 1/100 of the elution volume, was added to correct the elution. The elution was dialyzed for 24 h with PBS (pH 7.4) to remove salts.

### 2.9. Preparation of Gold Nanoparticles and Antibody Conjugates

Gold conjugate was used to label antigen-antibody binding in an easy-to-understand manner. First, a flask was filled with 1 L of tertiary distilled water and heated on a stirrer. When the distilled water began to boil at 100 °C, the mixture was treated with 100 mL of 10% trisodium citrate and 100 mL of 0.1% gold chloride and stirred with a magnetic bar. When the yellow solution turned purple, it was heated for an additional 10 min, transferred to a 1-L glass bottle, and cooled slowly. The gold nanoparticle solution was titrated at pH 7.5, using 1 M K_2_CO_3_, and conjugated with the antibodies based on the size of OD 2.0 at 540 nm absorbance. After stirring at 25 °C for 10 min, blocking was performed by adding 0.25% casein.

### 2.10. Rapid Diagnosis Kit Production and Components

A half-pair test was performed to examine reactivity between antibodies (Table 4). The recombinant NT-proBNP antigen was diluted in a gold nanoparticle condensation solution and dispensed at a concentration of 10 ng/mL into a 96-well plate. Positive and negative strips were prepared, and the two control lines were treated with goat anti-mouse IgG (1 mg/mL). The candidate clonal antibody was dispensed at 2 mg/mL in the positive strip test line. In contrast, 100 mM phosphate buffer (pH 7.4) was dispensed onto the negative strip test line to determine the presence or absence of false positives. Subsequently, antibodies with good sensitivity were selected using the half-pair test. Diagnostic kits were produced using candidate clonal antibodies to perform full-strip pair tests. A composition of the diagnostic kits used is shown in Figure 1. A test line and a control line were placed on a nitrocellulose membrane. Thereafter, 2 mg/mL antibody was dispensed onto the test line to capture the antigen, whereas 1 mg/mL goat anti-mouse IgG was dispensed onto the control line.

### 2.11. Antigen-Antibody Reactivity Test

The recombinant NT-proBNP antigen (HyTest, Turku, Finland) required for accurate antigen-antibody reactivity tests, as well as different concentrations of NT-proBNP plasma (ProMedex, Marlboro, NJ, USA) from four different patients, were purchased from the indicated companies (Table 5). For the test, 100 µL of the sample was dispensed onto the sample pad, and the reaction was examined after approximately 10 min. The control line was examined for all experiments. The appearance of a band on the test line was considered positive.

### 2.12. Statistical Analysis

Statistical analyses were performed using frequency analysis and graphing with MS Excel (Microsoft, Redmond, Washington, DC, USA) and SPSS Statistical Procedure for Windows (SPSS PASW Statistical 23.0; SPSS Inc., Chicago, IL, USA). Cohen’s kappa statistic was used for agreement on the measurement category values.

## 3. Results

### 3.1. NT-proBNP Gene Synthesis and Cloning

A primer containing HindIII and XhoI restriction enzyme cleavage sites was designed, using NT-proBNP as a template, by Bioneer (Korea) and used in the PCR. An analysis of the amplified NT-proBNP product showed a band of approximately 228 bp on a 1.5% agarose gel (Figure 2). The identified band was extracted with the desired target inserts using a gel extraction kit and recombined using a TA-cloning kit. Following the transformation with *E. coli* DH5α, subcloning was performed using the pET-21b expression vector to express the recombinant protein.

### 3.2. Recombinant Protein Expression and Purification

Proteins were extracted from transformed *E. coli* BL21 (DE3) and purified via Ni-NTA affinity chromatography. Analysis of the eluted protein via SDS-PAGE revealed a thick protein band of approximately 13 kDa (Figure 3). The recombinant NT-proBNP protein was used as an antigen to produce monoclonal antibodies for immunization.

### 3.3. Immunization and Hybridoma Screening

Hybridomas, produced by the fusion of lymph nodes from immunized mice with Sp2/O cells, were selectively cultured and screened via indirect ELISA (Table 3). After the screening, approximately 11 samples (C1E4, C1G3, C1H12, C2G11, C2H2, C3B11, C3G8, C3H2, C4B12, C4D6, and C4H8) showed high absorbance. Seven monoclonal candidates were established by serially diluting the selected clones several times (Table 4, Figure 4).

### 3.4. Mass Production and Purification of Monoclonal Antibodies

To obtain a large number of ascites and purify the antibodies, ascites were collected by injecting fused cells into the abdominal cavity of the mouse. Protein A+G affinity chromatography was used to bind ascites to the resin. Following adequate washing with PBS (pH 7.4), nonspecific substances were removed, and the antibody was eluted based on the difference in pH.

### 3.5. Monoclonal Antibody Pair Test

Half-pair tests were performed by selecting seven candidate clonal antibodies. Clonal antibodies were dispensed on the membrane at a concentration of 2 mg/mL, and gold nanoparticle conjugate solutions were prepared. To select gold nanoparticle-conjugated antibodies and capture antibodies, approximately 49 cases were substituted. As a result of the half-pair test, eight highly sensitive pairs were selected, and the kit was assembled and tested to accurately diagnose its reactivity. Seven kits were prepared per pair test, and serum (negative plasma) of healthy individuals was selected as a control to determine false positives. To examine the reactivity of the antigen, the recombinant NT-proBNP antigen developed in this study and recombinant NT-proBNP antigen purchased from HyTest were diluted and dispensed at 10 ng/mL. The NT-proBNP serum (plasma) purchased from ProMedex (100 µL each) was dispensed in the order of high to low concentrations (35, 25, 16, and 10 ng/mL). The pair test indicated that, among the eight pairs of tests, the best sensitivity was shown by the H:C3B11(Capture)/C4B12(Gold) pair (Figure 5).

## 4. Discussion

Globally, the HF rate has been increasing exponentially with the aging population. Increased life expectancy in the United States has resulted in one in three individuals suffering from some form of heart disease [18]. Heart failure, a progressive disease, is the most common cause of death due to cardiovascular ailments. Its prevalence is increasing along with the number of patients and elderly individuals with chronic heart diseases due to the availability of various treatment technologies. Thus, HF is considered an important disease in terms of socioeconomics [19,20].

Echocardiography is considered the gold standard for diagnosing HF, but its efficacy is compromised in patients with chronic lung disease and obese patients with respiratory failure [21,22]. Although various methods enabling the early detection of HF are available, these traditional methods do not ensure efficacy, accuracy, timeliness and cost-effectiveness of clinical decisions, prognoses, or treatments. In fact, delays in HF diagnosis can lead to inadequate treatment, increased medical expenses, extended hospital stays, and increased emergency room visits [23,24].

Current biomarkers for diagnosing HF include CRP and IL-6 inflammatory biomarkers, HF myocyte injury biomarkers CTnI and CK- MB, and HF myocyte stress biomarkers such as BNP and ST2. Among them, the specificity of BNP and ST2 is relatively higher than that of other markers. In particular, BNP is known to be a strong prognostic predictor for all causes of death in patients with HF and is released from the heart in response to hypertrophy, mechanical stress, increased myocardial tone under conditions of oxidative stress and increased intravascular volume. Therefore, BNP is routinely used to diagnose and monitor HF and is an important prognostic indicator of mortality, particularly in patients with HF [25,26]. Recently, NT-proBNP has been used as a routine laboratory test for diagnosing HF and has already proven to be useful in clinical tests for the differential diagnosis of acute respiratory distress, HF diagnosis and prognosis in several studies [27]. In this study, based on the NT-proBNP test system, a new NT-proBNP amino acid sequence was selected based on the NCBI Human proBNP database to develop an antibody for application in a rapid diagnostic kit in a clinical setting. The target gene was then amplified through PCR. In addition, a high-purity NT-proBNP recombinant antigen was obtained by subcloning, using a vector, and purifying the target protein. This antigen was used for immunization. A large number of monoclonal antibodies was produced through fusion cell selection and purification and was applied to an optimal rapid diagnostic kit to confirm reactivity with a commercially available standard antigen, that is, sensitivity and specificity. Thus, a new recombinant protein antigen based on human NT-proBNP was developed, and a large number of monoclonal antibodies was obtained using this antigen. In addition, the analysis of antigen-antibody reactivity by applying it to a gold conjugate-based optimal diagnostic kit revealed superior sensitivity and specificity. We plan on applying to related organizations for permission to commercialize once we have finished developing a reader machine with high interoperability using the diagnostic kit which applied this antibody protein and conduct additional clinical trials.

## 5. Conclusions

In conclusion, the field application of the CHF rapid diagnosis kit constructed using the recombinant NT-proBNP antigen developed in this study may reduce costs and replace methods requiring expensive equipment and expertise to diagnose related diseases. Thus, it may substantially contribute to the prognosis and treatment of heart diseases.

## Figures and Tables

**Figure 1 medicina-57-00751-f001:**
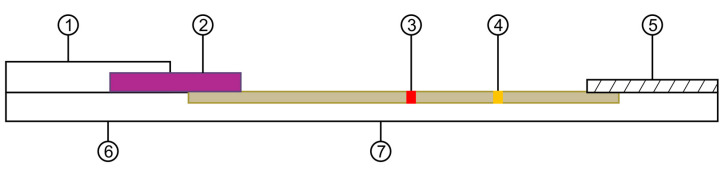
Structure of the rapid diagnosis kit. ① sample pad; ② conjugated pad; ③ test line; ④ control line; ⑤ absorbance pad; ⑥ plastic backbone; ⑦ nitrocellulose membrane.

**Figure 2 medicina-57-00751-f002:**
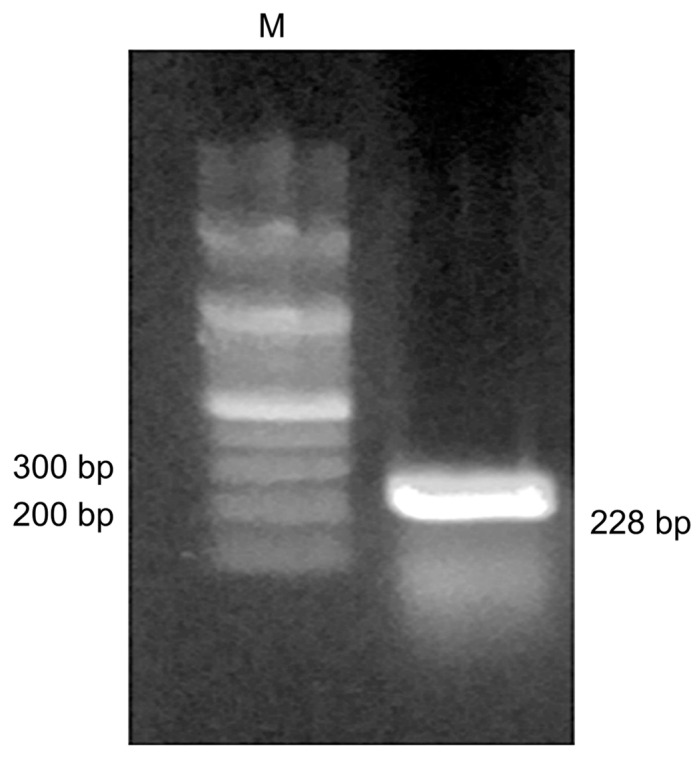
Agarose gel electrophoresis analysis of the NT-proBNP DNA fragment amplified via PCR. Lane M indicates the 100 bp DNA ladder. Lane L contains the PCR products of NT-proBNP. The approximate size of the amplified DNA fragment was 228 bp.

**Figure 3 medicina-57-00751-f003:**
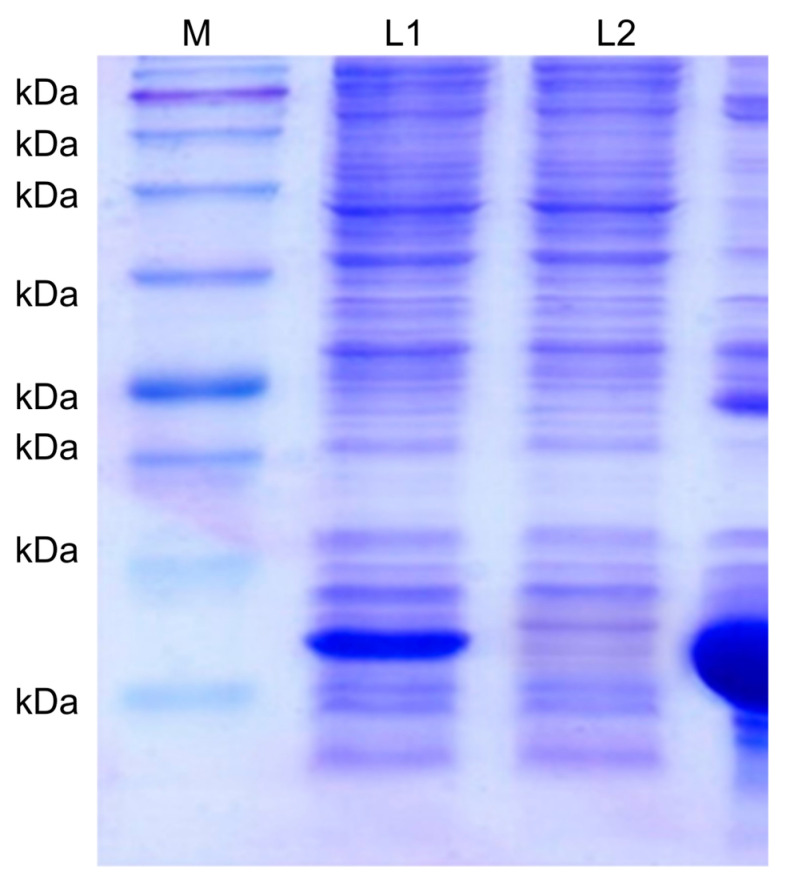
SDS-PAGE analysis of the NT-proBNP recombinant protein. M: protein marker (kDa); L1: supernatant (lysate); L2: filtrate; L3: elution.

**Figure 4 medicina-57-00751-f004:**
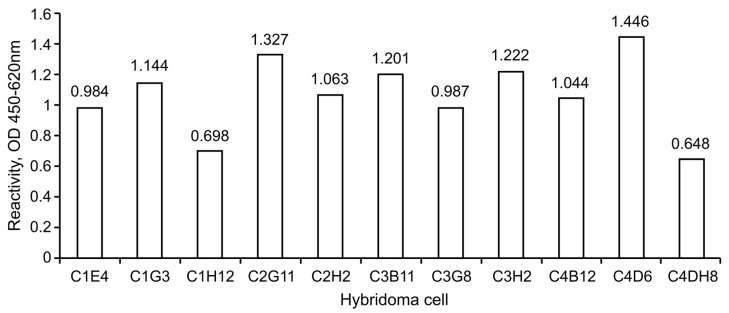
Serial dilution for monoclonal antibody screening. Final clones were selected through serial dilution: C1G3 (1.144), C2G11 (1.327), C2H2 (1.063), C3B11 (1.201), C3H2 (1.222), C4B12 (1.044), and C4D6 (1.446).

**Figure 5 medicina-57-00751-f005:**
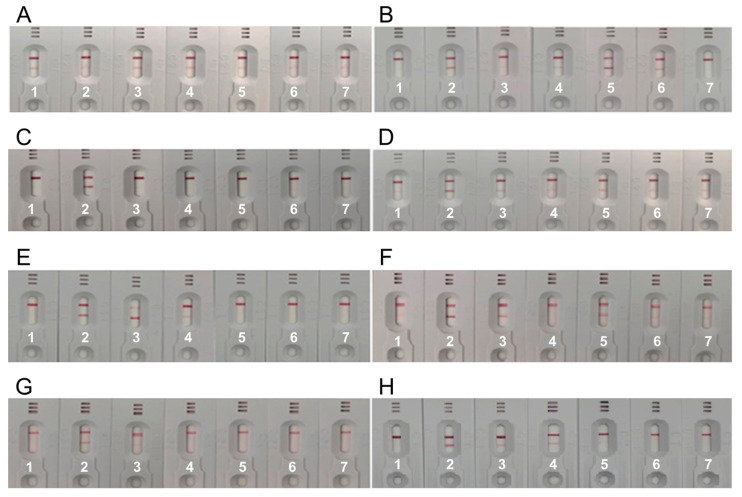
Specificity of the NT-proBNP rapid diagnosis kit. Letters “C” and “G” stand for Capture and Gold, respectively. (**A**) C2H2(C)/C1G3(G); (**B**) C3B11(C)/C1G3(G); (**C**) C1G3(C)/C2H2(G); (**D**) C3B11(C)/C2H2(G); (**E**) C4D6(C)/C2H2(G); (**F**) C3B11(C)/C3B11(G); (**G**) C2H2(C)/C4B12(G); (**H**) C3B11(C)/C4B12(G).

**Table 1 medicina-57-00751-t001:** NT-proBNP amino acid sequence.

Gene	Amino Acid Sequence
NT-proBNP	001-HPLGSPGSASDLETSGLQEQRNHLQGKLSE
	031-LQVEQTSLEPLQESPRPTGVWKSREVATEG
	061-IRGHRKMVLYTLRAPR

**Table 2 medicina-57-00751-t002:** PCR primer sequences for the amplification of NT-proBNP.

Gene		Amino Acid Sequence
NT-proBNP	Forward	5′- AAG CTT CAC CCG CTG GGC AGC CCC-3′
	Reverse	5′- CTC GAG TCG TGG TGC CCG CAG GGT-3′

The forward and reverse primers contain HindIII (AAGCTT) and XhoI (CTCGAG) restriction sites.

**Table 3 medicina-57-00751-t003:** Indirect ELISA for primary screening.

Plate 1	C1	1	2	3	4	5	6	7	8	9	10	11	12
	A	0.422	0.472	0.303	0.353	0.3	0.411	0.289	0.338	0.335	0.258	0.292	0.398
	B	0.281	0.274	0.313	0.304	0.247	0.76	0.263	0.275	0.284	0.275	0.258	0.295
	C	0.288	0.233	0.72	0.263	0.249	0.3	0.287	0.259	0.267	0.287	0.261	0.283
	D	0.348	0.26	0.29	0.265	0.287	0.263	0.258	0.277	0.585	0.222	0.693	0.303
	E	0.31	0.254	0.684	**1.043**	0.265	0.289	0.279	0.479	0.232	0.242	0.237	0.308
	F	0.37	0.225	0.268	0.272	0.305	0.451	0.275	0.302	0.307	0.267	0.294	0.278
	G	0.355	0.276	**1.197**	0.334	0.325	0.231	0.251	0.287	0.292	0.253	0.274	0.308
	H	0.325	0.333	0.337	0.303	0.366	0.347	0.27	0.308	0.264	0.279	0.282	**0.833**
Plate 2	C2	1	2	3	4	5	6	7	8	9	10	11	12
	A	0.307	0.309	0.276	0.27	0.332	0.342	0.281	0.252	0.297	0.299	0.298	0.358
	B	0.281	0.25	0.282	0.269	0.28	0.303	0.288	0.302	0.267	0.241	0.239	0.315
	C	0.454	0.293	0.272	0.302	0.281	0.27	0.258	0.253	0.259	0.27	0.218	0.342
	D	0.304	0.26	0.289	0.317	0.313	0.28	0.32	0.321	0.288	0.253	0.253	0.275
	E	0.3	0.255	0.334	0.287	0.289	0.282	0.298	0.278	0.295	0.279	0.298	0.25
	F	0.254	0.249	0.275	0.439	0.263	0.288	0.46	0.279	0.34	0.259	0.252	0.278
	G	0.311	0.359	343	0.278	0.31	0.29	0.381	0.286	0.347	0.287	**1.137**	0.465
	H	0.411	**0.984**	0.295	0.272	0.287	0.284	0.289	0.525	0.341	0.254	0.313	0.317
Plate 3	C3	1	2	3	4	5	6	7	8	9	10	11	12
	A	0.497	0.384	0.444	0.375	0.447	0.399	0.353	0.477	0.43	0.41	0.48	0.661
	B	0.364	0.371	0.356	0.364	0.368	0.312	0.339	0.378	0.369	0.385	**0.934**	0.41
	C	0.442	0.409	0.38	0.375	465	0.398	0.38	0.407	0.357	0.461	0.357	0.389
	D	0.427	0.353	0.404	0.382	0.403	0.404	0.455	0.388	0.383	0.392	0.364	0.367
	E	0.396	0.397	0.394	0.46	0.394	0.422	0.4	0.367	0.4	0.359	0.338	0.446
	F	0.418	0.352	0.401	0.422	0.379	0.408	0.4	0.436	0.385	0.33	0.319	0.406
	G	0.398	0.326	0.378	0.416	0.399	0.385	0.384	**0.916**	0.375	0.33	0.415	0.484
	H	0.348	**1.027**	0.296	0.247	0.28	0.297	0.275	0.306	0.248	0.264	0.293	0.242
Plate 4	C4	1	2	3	4	5	6	7	8	9	10	11	12
	A	0.301	0.265	0.324	0.371	0.435	0.44	0.387	0.402	0.387	0.356	0.365	0.397
	B	0.442	0.38	0.416	0.328	0.538	0.406	0.377	0.361	0.348	0.384	0.364	**0.912**
	C	0.403	0.391	0.407	0.451	0.359	353	0.438	0.555	0.401	0.411	0.394	602
	D	0.584	0.389	0.423	0.354	0.386	**1.339**	0.42	0.375	0.497	0.375	0.377	356
	E	0.425	0.385	0.395	0.407	0.413	416	0.371	0.363	0.418	0.424	0.304	412
	F	0.433	0.387	0.423	0.4	0.449	0.371	0.361	0.343	0.372	0.349	0.346	377
	G	0.383	0.332	0.28	0.383	0.378	0.333	0.379	0.381	0.321	0.341	0.325	425
	H	0.373	0.315	0.373	0.37	0.403	0.364	0.385	**0.843**	0.406	0.359	0.322	406

Bold font represents wells with high titers. The titers of red boxes are as follows: Plate 1, C1E4 (1.043), C1G3 (1.197), C1H12 (0.833); Plate 2, C2G11 (1.137), C2H2 (0.984); Plate 3, C3B11 (0.934), C3G8 (0.916), C3H2 (1.027); Plate 4, C4B12 (0.912), C4D6 (1.339), C4H8 (0.843).

**Table 4 medicina-57-00751-t004:** Results of the half-pair test.

G/C	C1G3	C2G11	C2H2	C3B11	C3H2	C4B12	C4D6
C1G3	−	1+	3+	3+	2+	−	−
C2G11	1+	−	−	2+	1+	1+	1+
C2H2	−	−	−	3+	−	2+	3+
C3B11	2+	1+	−	3+	−	1+	2+
C3H2	2+	−	−	2+	−	1+	1+
C4B12	−	1+	3+	3+	2+	−	−
C4D6	−	1+	2+	1+	1+	−	−

Specific sensitivity: 3+ (strong positive); 2+ (weak positive); 1+ (positive); − (negative).

**Table 5 medicina-57-00751-t005:** NT-proBNP plasma specimens.

Lane	Specimen	Concentration (ng/mL)	Sex	Age (Years)
(1)	Normal serum	-	-	-
(2)	Recombinant Antigen	10	-	-
(3)	Commercial Antigen	10	-	-
(4)	Patient’s serum ⓵	35	Female	59
(5)	Patient’s serum ⓶	25	Female	93
(6)	Patient’s serum ⓷	16	Female	86
(7)	Patient’s serum ⓸	10	Female	91

## Data Availability

The datasets used and/or analysed during the current study are available from the corresponding author on reasonable request.

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
