# Peer review of "Development of a Rapid Diagnostic Kit for Congestive Heart Failure Using Recombinant NT-proBNP Antigen"

_medicina, 2021, doi:10.3390/medicina57080751_

Round 1

Reviewer 1 Report

Dear author,

about the submitted manuscript, enclosed my suggestions:

INTRODUCTION:

Please report the importance of BNP as predictor of survival in patients with heart failure (HF),  and also of other stretch, injury and inflammation markers to predict clinical outcomes in heart failure (HF) patients and in those at higher risk treated with implantable cardioverter defibrillator (ICD), (Front Physiol. 2018 Jun 26;9:758. doi: 10.3389/fphys.2018.00758).

METHODS:

Please include the description of:

-Inclusion and exclusion study criteria;

-study population sample size;

-Study beginning and ending, and follow-up duration;

-I do not see a Statistical analysis paragraph.

RESULTS:

Are You studying a population of patients? I do not understand. Is this a clinical study? If yes, please give us number of study population.

What is the percentage of patients treated with ICD and cardiac resynchronization therapy? The CRTd could reduce the HF hospitalization and deaths, and this could result in the significant reduction of BNP and other stretching biomarkers (Cardiovasc Diabetol. 2017 Jun 9;16(1):75. doi: 10.1186/s12933-017-0554-2). Please discuss this point and this reference in the text.

What was the percentage of type 2 diabetes mellitus (T2DM) in the study population? T2DM could negatively affect clinical outcomes in HF patients, and alter the blood values of BNP at baseline and after medical and interventional therpaies (Cardiovasc Diabetol. 2020 Nov 28;19(1):202. doi: 10.1186/s12933-020-01180-8). Please discuss it.

DISCUSSION:

Please short it, it is too long.

Please give to readers a clinical message.

Improve English quality of the text.

Improve quality of figures and tables.

Author Response

Document attached below.

Reviewer 2 Report

The manuscript is complete and interesting. the possibility of using a simple kit for the analysis of one of the strongest markers from a diagnostic and prognostic point of view in heart failure is fundamental for the management of patients with this diagnosis and to discriminate, for example, those who have symptoms and still a dagnosis correct has been placed.

the data in the manuscript are very detailed and easily legible. it is a good and useful job in clinical practice and in areas of development of such methods.

Author Response

Document attached below.

Reviewer 3 Report

Thank you for the opportunity to review this ms. on the engineering of NT-antibodies detecting NT-proBNP. 

I am sorry to say that I am not able to judge the soundness of the methodology described in this ms. Please let me give some comments:

1) As far as I know, Roche diagnostics has a patent on the measurement of circulating NT-proBNP. May there be a violation of patient rights?

2) It is interesting to learn that the authors succeeded to construct antigens and raise antibodies against parts of the NT-proBNP protein. However, it is not clear for me whether this antibody has sufficient specificity to measure NT-proBNP in patients. This is a laboratory study and it is unclear whether the results presented are relevant for clinical use.

3) The length of the ms. could be considerably reduced. In addition, English language should be corrected by an English native.

Author Response

Document attached below.

Round 2

Reviewer 1 Report

Dear authors,

I see improvement in the current revised manuscript. By the way, I have just few comments:

-The statistical analysis paragraph is not complete. Indeed, you have to indicate the test used for comparison between study variables (T trest, Fisher test etc.), and much more indicate the p<0.05 as statistical significant p value. Correct it and add thius information in the text.

-Please remark the evaluation of BNP in patients with heart failure (HF) and advanced HF as those receiving CRTd device (NYHA class II/III and refractoriness to anti-HF medical therapies). Indeed, in these patients the BNP is a diagnostic marker, that could be modulated by anti-HF therapies (Cardiovasc Diabetol. 2017 Jun 9;16(1):75. doi: 10.1186/s12933-017-0554-2; Cardiovasc Diabetol. 2018 Oct 22;17(1):137. doi: 10.1186/s12933-018-0778-9), and predict clinical outcomes ( Cardiovasc Diabetol. 2020 Nov 28;19(1):202. doi: 10.1186/s12933-020-01180-8).

Please remark these concepts and add the suggested references to the current manuscript.

Author Response

Responses to the comments of Reviewer 1

Comments and Suggestions for Authors

Dear authors,

I see improvement in the current revised manuscript. By the way, I have just few comments:

Q1) The statistical analysis paragraph is not complete. Indeed, you have to indicate the test used for comparison between study variables (T trest, Fisher test etc.), and much more indicate the p<0.05 as statistical significant p value. Correct it and add thius information in the text.

Response: This study is a laboratory study investigating the feasibility of developing raw materials. That is, after developing the NT-proBNP-related antigen and antibody, it was applied to an optimal diagnostic kit to confirm specificity and sensitivity compared with those of commercially available standard antigens. Laboratory studies do not mention differences in means and significance levels between diagnoses, as there is no reason to compare the mean values between groups, unlike in clinical trials. In the future, as an additional experiment, all possible statistical techniques, such as covariance analysis, differences between populations, and significance levels will be applied at the clinical trial stage to develop and commercialize an independent machine linked to the diagnostic kit.

Q2) Please remark the evaluation of BNP in patients with heart failure (HF) and advanced HF as those receiving CRTd device (NYHA class II/III and refractoriness to anti-HF medical therapies). Indeed, in these patients the BNP is a diagnostic marker, that could be modulated by anti-HF therapies (Cardiovasc Diabetol. 2017 Jun 9;16(1):75. doi: 10.1186/s12933-017-0554-2; Cardiovasc Diabetol. 2018 Oct 22;17(1):137. doi: 10.1186/s12933-018-0778-9), and predict clinical outcomes (Cardiovasc Diabetol. 2020 Nov 28;19(1):202. doi: 10.1186/s12933-020-01180-8).

Please remark these concepts and add the suggested references to the current manuscript.

Response: Thank you for these great points and advice. BNP evaluations in patients with heart failure (HF) and patients with advanced HF receiving CRTd devices have been described and references have been added. (Page 2, Lines 69–74) (Page 12, Lines 410–421)

Cardiac resynchronization therapy (CRT) is highly effective for heart failure patients with type 2 diabetes mellitus, significantly improving clinical symptoms and reducing mortality and long-term morbidity [11,12]. A wide range of biomarkers can be used as predictors of the CRT response in heart failure patients receiving such treatments. Among them, BNT-proBNP has high sensitivity and specificity, and there is a significant correlation between BNP levels and CRT results [13,14].

11.    Sardu, C.; Paolisso, P.; Sacra, C.; Santamaria, M.; Lucia, C.; Ruocco, A.; et al. Cardiac resynchronization therapy with a defibrillator (CRTd) in failing heart patients with type 2 diabetes mellitus and treated by glucagon-like peptide 1 receptor agonists (GLP-1 RA) therapy vs. conventional hypoglycemic drugs: arrhythmic burden, hospitalizations for heart failure, and CRTd responders rate. Cardiovasc Diabetol. 2018, 17, 137. DOI: 10.1186/s12933-018-0778-9.

12.    Sardu, C.; Paolisso, P.; Ducceschi, V.; Santamaria, M.; Sacra, C.; Cardiac resynchronization therapy and its effects in patients with type 2 DIAbetes mellitus OPTimized in automatic vs. echo guided approach. Data from the DIA-OPTA investigators. Cardiovasc Diabetol. 2020, 202, DOI: 3/s12936) 020-01180-8.

13.    Debska-Kozlowska, A.; Ksiazczyk, M.; Warchol, I.; Lubiski, A. Clinical usefulness of N-terminal prohormone of brain natriuretic peptide and high sensitivity troponin T in patients with heart failure undergoing cardiac resynchronization therapy. Curr Pharm Des. 2019, 25,1671–1678. DOI: 10.2174/1381612825666190621155718.

14.    Asgardoon, M.H.; Vasheghani-Farahani, A.; Sherafati, A. Usefulness of biomarkers for predicting response to cardiac resynchronization therapy. Curr Cardiol Rev. 2020, 16 132–140. DOI: 10.2174/1573403X15666191206163846.

Reviewer 3 Report

Thank you very much to review the revised ms. on the Development of a rapid diagnostic kit for congestive heart failure using recombinant NT-proBNP antigen. 

The authors have addressed my comments. Nevertheless, I am not experienced to judge the validity of their presentation. In addition, I would strongly suggest to submit this paper in a Biochemical or Laboratory Journal. 

Author Response

Responses to the comments of Reviewer 3

Comments and Suggestions for Authors

Q1) Thank you very much to review the revised ms. on the Development of a rapid diagnostic kit for congestive heart failure using recombinant NT-proBNP antigen. 

The authors have addressed my comments. Nevertheless, I am not experienced to judge the validity of their presentation. In addition, I would strongly suggest to submit this paper in a Biochemical or Laboratory Journal. 

Response: First, thank you very much for your good comments and suggestions. This paper took a long time to complete. Basically, the main purpose is to commercialize this approach, but after consultation with a partner company, we submitted the content to this journal. Submission of additional related papers, which will be generated in the future, to the proposed type of journal will be carefully considered. However, we ask that you consider the fact that we have submitted this paper because we think that papers related to the BNP diagnostic kit are also suitable for this journal. Thank you very much.